# The “Near”-Narrowed Internal Auditory Canal Syndrome in Adults: Clinical Aspects, Audio-Vestibular Findings, and Radiological Criteria for Diagnosis

**DOI:** 10.3390/jcm12247580

**Published:** 2023-12-08

**Authors:** Eugen C. Ionescu, Pierre Reynard, Samar A. Idriss, Aicha Ltaief-Boudriga, Charles-Alexandre Joly, Hung Thai-Van

**Affiliations:** 1Department of Audiology and Neurotology, Civil Hospitals of Lyon, 69003 Lyon, France; pierre.reynard@chu-lyon.fr (P.R.); charles-alexandre.joly01@chu-lyon.fr (C.-A.J.); hung.thai-van@chu-lyon.fr (H.T.-V.); 2Team Clinical and Translational Exploration of Sensorineural Hearing Loss, Hearing Institute, Research Center of Pasteur Institute, Inserm U1120, 75015 Paris, France; 3Department of Physiology, Claude Bernard University, 69003 Lyon, France; 4Department of Otolaryngology, Dar Al Shifa Hospital, Hawally 13034, Kuwait; 5Department of Otolaryngology and Head and Neck Surgery, Eye and Ear Hospital, Holy Spirit University of Kaslik, Beirut 1201, Lebanon; 6Department of Radiology, Civil Hospitals of Lyon, 69003 Lyon, France; aicha.ltaief-boudrigua@chu-lyon.fr

**Keywords:** narrowed internal auditory canal syndrome, vestibular paroxysmia, cross compression syndrome, sodium channel blockers

## Abstract

Introduction: Vestibular Paroxysmia (VP) refers to short attacks of vertigo, spontaneous or triggered by head movements, and implies the presence of a compressive vascular loop in contact with the cochleovestibular nerve (CVN). Classically, a narrowed internal auditory canal (IAC) corresponds to a diameter of less than 2 mm on CT, usually associated with a hypoplastic CVN on MRI. The aim of this study was to discuss a distinct clinical entity mimicking VP in relation to a “near”-narrowed IAC (NNIAC) and to propose radiological criteria for its diagnosis. Methods: Radiological measurements of the IAC were compared between three groups: the study group (SG, subjects with a clinical presentation suggestive of VP, but whose MRI of the inner ear and pontocerebellar angle excluded a compressive vascular loop) and two control groups (adult and children) with normal vestibular evaluations and no history of vertigo. Results: 59 subjects (18 M and 41 F) were included in the SG. The main symptoms of NNIAC were positional vertigo, exercise- or rapid head movements-induced vertigo, and dizziness. The statistical analysis in the study group showed that the threshold values for diagnosis were 3.3 mm (in tomodensitometry) and 2.9 mm (in MRI) in coronal sections of IAC. Although a significantly lower mean value for axial IAC diameter was found in SG compared with controls, the statistics did not reveal a threshold due to the large inter-individual variations in IAC measurements in normal subjects. There was no significant difference in IAC diameter between the adult and pediatric controls. Conclusions: In the present study, we report a new anatomopathological condition that appears to be responsible for a clinical picture very similar—but not identical—to VP in association with the presence of an NNIAC. The diagnosis requires a careful analysis of the IAC’s shape and diameters in both axial and coronal planes.

## 1. Introduction

The cochleovestibular nerve (CVN) was first described as a possible origin of disabling vertigo by Gowers in 1877 [1]. In 1975, based on intraoperative observation, Jannetta et al. hypothesized a relationship between “disabling positional vertigo” and the presence of a blood vessel compressing the CVN in the Obersteiner–Redlich zone (or root entry zone (REZ)) [2,3]. This condition was defined as Neurovascular Cross Compression (NVCC) of the CVN. Shortly afterward, the term Vestibular Paroxysmia (VP) was introduced by Brandt et al. and corresponds to short attacks of spinning or non-spinning vertigo, lasting from a few seconds to a few minutes, triggered by head movements or changes of position [4]. The attacks might be accompanied by hyperacusis or typewriter-like tinnitus, but the most common accompanying symptom is unsteadiness [5,6]. Recently, the Barany Society has proposed criteria for definite and probable VP diagnosis. Anatomopathological VP classically refers to the presence of a vascular structure in the cerebellopontine angle (CPA) in contact with the CVN, causing segmental pressure-induced neuropathy [7]. Symptoms are generated by direct vascular pulsations in contact with CVN, resulting in ephapse [8]. This localized neuropathy generates discharges on the nerve, which can be excitatory or inhibitory for the concerned nerve fibers [4]. By analogy with trigeminal neuralgia, hemifacial spasm, glossopharyngeal neuralgia, or superior oblique muscle myokymia, the presence of a vessel partially compressing the CVN leads to a demyelination process and ephaptic discharges inducing brief vestibular and/or auditory symptoms [6]. Also, CVN can be injured by local compression in the transition zone, distant from the brainstem and without overlapping the REZ [9]. In these cases, compression may be generated by the anterior–inferior cerebellar artery, postero-inferior cerebellar artery, vertebral artery, or veins [9]. More rarely, similar clinical presentations were reported in relation to osteoma or exostosis of the internal auditory canal (IAC) [10,11,12].

A vestibular hypofunction during head impulse and/or caloric test was observed in 20–45% of VP patients [6,13]. The hyperventilation maneuver may induce an ipsi- or contralateral nystagmus [6,14,15]. High-resolution magnetic resonance imaging (MRI) with CISS/FIESTA sequences of the brainstem may support the diagnosis [6,7,16]. In fact, MRI not only visualizes the neurovascular contact itself but also the absence of cerebro-spinal fluid between the vessel and the nerve, which can be considered an indirect sign of contact [16]. Best et al. reported that the NVCC of the CVN was present in all VP patients with VP in a small sample (*n* = 20) but also in 7 controls out of 20 NVCC [13]. Furthermore, a cross compression can be found bilaterally on MRI in up to 42% of patients presenting with typical VP [6]. 

While microvascular decompression can improve VP [17], medical treatment with antiepileptic drugs such as carbamazepine was recommended by Brandt [4]. Other authors also suggested the interest of low-dose oxcarbazepine in disabling “typewriter tinnitus” induced by vascular compression of the cochlear nerve [5] as well as recurrent attacks of tinnitus and vertigo in NVCC of the CVN [18]. Patients who do not tolerate medication but respond to it if the affected side is clearly identified may be candidates for surgical decompression [4,18]. 

In our tertiary center, over the past few years, we had the opportunity to encounter patients with chronic vestibulopathy presenting a clinical profile quite similar to VP or auditory equivalents (e.g., accentuated tinnitus-type writing or triggered by certain stereotypical head movements and/or physical exercise more or less associated with pain or painful pressure in the external auditory canal). These patients were generally referred with a presumptive clinical diagnosis of possible Menière’s disease with an atypical clinical presentation, and many of them were also subject to gait instability corresponding to some extent to the published criteria for persistent postural perceptual vertigo (PPPD) [19]. In a large proportion of cases, the diagnosis was not obvious, and MRI was not in favor of significant NVCC to support the VP diagnosis. We therefore wondered whether there was any other identifiable anatomical feature in the CVN pathway, apart from a “classic” vascular compression, that might explain the clinical manifestations. For this purpose, we retrospectively sought out all cases presenting with these clinical features and carefully examined their MRI and high-resolution computed tomography (HRCT) images with morphometric analysis of the IAC since in some of these cases, as in some of these cases, this structure appeared narrowed. Similar symptoms and radiological signs have already been observed and reported in a pediatric series in which the initial presumptive diagnosis was either Benign Paroxysmal Vertigo of Childhood or VP by NVCC of the CVN [20]. Therefore, the aim of the present study is two-fold: (1) to discuss this apparently distinct form of CVN impairment in relation to a narrowed IAC, different from other pathological entities in the adult population, and (2) to propose radiological criteria for its diagnosis. Data regarding the smallest IAC diameters in the axial and coronal plane, obtained by MRI and HRCT imaging, were compared and discussed in three groups constituted according to the criteria set out below.

## 2. Materials and Methods

### 2.1. Population 

The inclusion criteria for the study group (SG) were (a) adults with clinical manifestations that may correspond to, are close to or may suggest probable VP as defined by the Barany Society [7]; (b) normal morphometric MRI appearance of the CVN in the IAC and ponto-cerebellar angle. Exclusion criteria were (a) subjects with otoneurological symptoms and findings consistent with another vestibular pathology (e.g., vestibular neuronitis, Menière’s disease, benign paroxysmal positional vertigo (BPPV)), (b) subjects with VP and strong radiological and clinical criteria in favor of NVCC and, (c) inner ear abnormalities/malformations such as the enlarged vestibular aqueduct, small semicircular canals or similar and hypoplastic CVN, among others. 

Two control groups homogeneous in number and sex distribution with SG, one of children and another of adults, all cochlear implanted patients, were also included in the radiological study. All subjects from the two control groups were fully explored by imaging by HRCT and MRI of the petrous bone and inner ear, these being strictly normal. The aim was to retrospectively verify if there are significant statistical variations between the measurements of the internal auditory canal made in all cases by MRI and, in most cases, also by HRCT between the three groups. The investigation adhered to the principles of the Declaration of Helsinki. Written and oral informed consent was obtained from the subjects.

### 2.2. Audio-Vestibular Assessment

All participants underwent thorough audio-vestibular evaluation. A detailed medical and otological history was taken, and a complete neurotological examination was performed by the same physician. Otoscopy and tympanometry were verified in each subject. As part of an auditory evaluation, each participant had a pure tone audiometry (PTA); when VP was suspected, auditory-evoked brainstem responses (ABR) were performed. According to Moller’s criteria [21,22], compression on the CVN induces a slowing of signal transmission by the respective auditory nerve fibers, and the CVN involvement was suspected as soon as the I-III interpeak latency (IPL) was prolonged more than 2.3 ms on the affected side.

All patients suspected of VP underwent a videonystagmoscopy to document the presence of spontaneous nystagmus and/or positional nystagmus. A full otoneurological work-up, including a fistula test and Valsalva maneuver, was performed, as described by Hain [23]. The Tullio phenomenon was excluded as the nystagmus was not triggered by loud acoustic stimulation (>100 dB HL) to both (bare) ears. The vestibular assessment in all groups included a videonystagmography (VNG, Ulmer^®^, Synapsis SA, Marseille, France) with 30 s bilateral caloric water stimulation, a VOR gain on rotary chair test, as well as a screening for induced, accentuated, or inhibited nystagmus on 30 s hyperventilation test as described earlier [6,14,15,24]. A video head impulse test (VHIT, Otometrics^®^, Taastrup, Denmark) was performed, too, with at least 10 strong head accelerations in the planes of each semicircular canal. Each semicircular canal is sensitive to endolymph displacement according to its specific anatomic orientation and, hence, to acceleration in that plane (see Rabbitt’s mathematical model [25]). A normal response was defined by a gain value above 0.8.

Saccular otolithic function was assessed on cVEMPs (Bio-Logic^®^ Nav-Pro system; stimulation by air conduction at 750 Hz tone bursts stimulation) during sternocleidomastoid contraction, and utricular function on oVEMPs (500 Hz tone bursts) during oblique inferior oculomotor muscle contraction. For the cVEMPs, latencies, and amplitudes of the first positive–negative peaks (p13–n23) were defined as described by Fujimoto et al. [26]. An absent response was defined by a non-reproducible p13–n23 over two attempts, and a diminished response was defined by a reproducible p13–n23 with an asymmetry ratio above the normal upper limits. 

### 2.3. Radiological Assessment

All subjects underwent a cerebral and inner ear 1.5 and/or 3 Tesla MRI to eliminate central pathologies and check for a CVN abnormality as described in previous studies [19,27]. Images of IAC in three planes showed morphometrically intact facial, vestibular, and cochlear nerves. The CVN course in the CPA and IAC was also closely analyzed to identify images suggestive of NVCC. When a vascular loop was confirmed, and the IAC’s diameter measurements were normal in both axial and coronal planes, no further imaging was requested, and the subject was excluded. When narrowing of the IAC was suspected, almost all subjects underwent HRCT of temporal bones with measurements of its length, axial and vertical diameter (VD) of the fundus, and axial and VD at the internal acoustic meatus [19,28]. In addition to the above conventional measurements, axial and coronal measurements were also taken along the axis of the IAC in the area where the canal appeared narrowest. Some patients of the SG (11/59) refused an additional evaluation by HRCT for personal reasons. As defined by Marques et al. in 110 normal subjects aged 1 to 92 years, the CT-normative data for the IAC were: length = 11.17 mm, VD fundus = 4.82 mm, and VD internal auditory meatus = 7.53 mm [29]. A narrowed internal auditory canal was suspected if the anteroposterior and/or coronal diameter was measured less than 3 mm [29,30]. All radiological analyses and measurements were carried out by the same radiologist (ALB) and double-verified by a senior neurotologist. We use Intellispace Portal Softwear version 12 (Philips Ingenia, Philips Healthcare, Amsterdam, The Netherlands) for all image post-treatment.

### 2.4. HRCT Steps for Assessing the Morphology and Dimensions of IACs

-Evaluation of the smallest anteroposterior and craniocaudal IAC diameters, both in the axial and coronal planes, after measuring the IAC length (Figure 1(I,II));-Description of bony abnormalities of the IAC walls (normal bone, fibrous dysplasia, meningeal calcifications, and/or osteoma of exostosis—if applicable);-Evaluation of any significant angulation or deformation of the IAC in the craniocaudal and anteroposterior planes.

### 2.5. MRI Steps for Assessing the Morphology of IACs

-Assessment of the perineural fluid environment in the IAC (particularly when contact between the CVN and IAC walls is visualized);-Assessment of any angulation between the cisternal pathway and intramedullary path of the CVN (high-resolution T2) or a significant angulation of the CVN in the IAC or at its meatus;-Evaluating the presence of a vascular structure compressing the CVN (characterized by neuronal indentation or deviation of the CVN pathway in the REZ or toward the origin of CVN in the brainstem);-Analysis of fusion images between high-resolution T2 and HRCT of the temporal bones (to improve assessment of the CVN trajectory and, therefore, detection of any deviations within the IAC in the axial or coronal plane);-Evaluation of anteroposterior and craniocaudal diameters of the IAC (Figure 2(I,II)).

### 2.6. Group’s Selection for the Radiological Comparative Morphometric Study

As previous publications showed that normal IAC dimensions do not significatively change with age [31], we retrospectively compared the study group (SG) with two control groups (CG1 and CG2, adults and children, respectively), although these subjects presented with profound deafness and were candidates for cochlear implants. For this reason, radiological measurements of the IAC were compared between three groups as follows: -The study group (SG) 59 subjects 18 M, 41 F) with a clinical presentation and audio-vestibular signs suggestive of defined or probable VP according to the Barany Society recommendations, but for whom attentive MRI assessment excluded a vascular loop or any other known retro cochlear pathologies;-Control group 1 (CG1): 59 adult (18 M, 41 F) subjects who received a cochlear implant (CI), having all normal inner ear HRCT and RMI imagery, pre- and post-operative vestibular evaluations, and who never reported vertigo before and after CI surgery;-Control group 2 (CG2): 59 children and adolescent subjects (18 M, 41 F) who received a CI, having all normal inner ear HRCT and RMI imagery, pre- and post-operative vestibular evaluations, and who never reported vertigo before and after CI surgery.

### 2.7. Data Processing and Analysis

An average of the right and left IAC measurements was documented for each participant (HRCT in IAC axial and coronal planes, MRI in IAC axial and coronal planes).

As not all distributions were normal, group means for each measure were compared using a non-parametric test. The three groups were compared using Kruskal–Wallis tests, and post hoc comparisons consisted of Wilcoxon Mann–Whitney tests with Bonferroni corrections. For each group, we also estimated the percentage of subjects whose measurement was less than or equal to various values ranging from 2.5 mm to 4 mm.

## 3. Results

### 3.1. Epidemiology

Fifty-nine patients were included in the SG (18 M, 41 F). Of these 59 subjects, 48 obtained HRCT images of sufficient quality. The mean age in the SG was 48.71 years (+/−17.8), with extreme ages being 18 and 88 years. Diagnosis of this pathology peaked in the fifth decade of life (Figure 3).

### 3.2. Clinical Findings

Symptoms and clinical signs for SG are presented in Table 1 and Table 2. Positional vertigo of short duration without latency (*n* = 39), dizziness accentuated by head movements (*n* = 26), and chronic imbalance and dizziness (*n* = 23) were the most frequently reported symptoms by subjects within the study group. Unilateral (*n* = 1) or bilateral (*n* = 8) constant tinnitus, unilateral (*n* = 5) or bilateral (*n* = 15) exercise-induced tinnitus, unilateral (*n* = 5) or bilateral (*n* = 17) hyperacusis, and unilateral (*n* = 4) or bilateral (*n* = 7) progressive sensorineural hearing loss were also observed, albeit with a lower incidence. Ten patients complained of otalgia or ear fullness.

Prior to their consultation at our tertiary center, patients of the study group were diagnosed as having either PPPD, recurrent bilateral vestibulopathy, Menière’s disease, or vestibular neuronitis. A total of 29 out of 59 subjects had no or only a poor response to acetylleucine treatment.

### 3.3. Audio-Vestibular Findings 

VNG, VHIT, or cVEMPs/oVEMPs were not specific for diagnosis, even when performed in a period of clinical recrudescence. However, although not indicative of the affected side, spontaneous nystagmus was observed in 8/59 patients (5 ipsilateral to the most radiologically narrowed IAC, 3 contralateral). The hyperventilation test was positive in 14/59 subjects (in nine cases, ipsilateral to the most radiologically narrowed IAC, five contralateral).

ABR evaluation was performed in 44 of the 59 patients. Pathological findings with a retrocochlear profile according to Moller’s criteria were documented in 24 subjects (13 bilateral and 11 unilateral). In contrast, ABR findings were normal bilaterally in 20 patients. In 15 patients, ABR was not performed since their PTA and intelligibility were strictly normal for their age. 

### 3.4. Radiological Findings: Group Comparison

In the study group (SG) subjects, IAC’s diameters were found to be significantly smaller than the subjects from the two control groups (Figure 3). No significant differences were found between the two control groups’ (CG1 and CG2) measurements. For axial HRCT measurements, the Kruskal–Wallis test revealed significant differences between groups (chi-squared = 29.648, df = 2, *p*-value < 0.001). Wilcoxon post hoc tests revealed significant differences between SG and CG1 (*p* < 0.001) and CG2 (*p* < 0.001) but not between CG1 and CG2 (*p* = 0.298) (Figure 4A). For coronal HRCT measurements, the Kruskal–Wallis test revealed significant differences between groups (chi-squared = 88.242, df = 2, *p*-value < 0.001). Wilcoxon post hoc tests revealed significant differences between the SG and the CG1 (*p* < 0.001) and CG2 (*p* < 0.001) but not between CG1 and CG2 (*p* = 1) (Figure 4B). For axial MRI measurements, the Kruskal–Wallis test revealed significant differences between groups (chi-squared = 35.027, df = 2, *p*-value < 0.001). Wilcoxon post hoc tests revealed significant differences between the SG and the CG1 (*p* < 0.001) and the CG2 (*p* < 0.001) but not between CG1 and CG2 (*p* = 0.063) (Figure 4C). For coronal MRI measurements, the Kruskal–Wallis test revealed significant differences between groups (chi-squared = 97.678, df = 2, *p*-value < 0.001). Wilcoxon post hoc tests revealed significant differences between SG and CG1 (*p* < 0.001) and CG2 (*p* < 0.001) but not between CG1 and CG2 (*p* = 1) (Figure 4D).

### 3.5. Subject Repartition according to the IAC Dimensions

For each group and each measurement, the percentage of effective CAI dimensions less than or equal to various values ranging from 2.5 mm to 4 mm is shown in Figure 5.

## 4. Discussion

In this study, we report a clinical entity comparable to paroxysmal vertigo in the absence of CVN NVCC in the IAC but in the presence of a small IAC diameter in the axial plane and/or especially in the coronal plane of the HRCT. Furthermore, there appeared to be no significant age-related difference in IAC diameter between CG1 and CG2 (Figure 4 and Figure 5). These results are in line with previously reported data supporting the hypothesis that IAC dimensions do not vary significantly over the lifespan [31].

Although axial slices appear less discriminating than coronal slices, it seems possible to retain the cut-off values for coronal slices of 3.3 mm (in HRCT) and 2.9 mm (in MRI) (corresponding to less than 10% false positives and less than 10% false negatives). We also wanted to assess whether IAC dimensions could help diagnose the clinical entity described here. To do this, we calculated for each plane (axial HRCT, coronal HRCT, axial MRI, and coronal MRI) and for each group (SG, CG1, and CG2) the proportion of subjects falling below different threshold values ranging from 2.5 mm to 4 mm (Figure 5). If the differences between SG and CG1-2 seemed too small to be discriminating in the axial plane, measurements taken in the coronal plane seem more appropriate to propose an anatomical criterion for our entity. Indeed, observation of the cumulative number of sub-objects in the different groups shows that 90% of the SG have coronal dimensions of less than 3.3 mm on HRCT and 2.9 mm on MRI, while 90% of CG1-2 subjects have dimensions greater than these values. Taking these values as anatomical thresholds, we would thus have, in our dataset, less than 10% false positives for SG and 10% false negatives for CG1-2. Furthermore, according to the CT cut-off values, the IACs appeared bilaterally narrowed in all but one patient (58 subjects, 98.3%). In light of these radiological results, we can observe that the diagnosis of narrowed IAC can also be supported by exclusive 3 T MRI evaluation of the IAC in coronal planes, as occurred in this series in 11 subjects. In these patients, the clinical presentation and the radiologic measurements of an IAC (smaller than 2.9 mm) were sufficient for the subsequent therapeutic approach, even if an HRCT was not performed. Suitable-quality MRI allows estimating not only the size, morphology, and course of the CVN in the IAC but also the presence of any vascular contact or other obstacles with a possibly compressive effect on the CVN or other local anatomical conditions that could lead to similar symptoms.

### 4.1. The Near-Narrowed Internal Auditory Canal: A Neglected Cause of VP and/or Assimilated Clinical Condition

Valvassori first reported the normal dimensions of the IAC [32,33]. A diameter of less than 2 mm in the axial plane is the most accepted criterion for evoking a narrowed IAC [34,35,36,37,38,39]. Since a narrow IAC implies hypoplastic or absent CVN, we suggest the notion of a “nearly narrowed IAC” (NNIAC). This means that even if the diameter of the IAC is below standard values—i.e., less than 3.5 mm—the CVN retains its morphometric characteristics on MRI. The diagnosis of VP can usually be suspected even in the absence of a typical MRI image of contact between a vascular loop and the CVN [40]. We believe that an NNIAC can be the cause of an audio-vestibular pathology that appears quite similar or very close to VP, as defined by the Barany Society [7]. Therefore, we can speculate that some successfully diagnosed and treated VP, without radiologic evidence of an NVCC, would, in fact, be related to an NNIAC.

Similarly, in a recent study, our team previously reported that NNIAC could generate VP-evocative symptoms in a cohort of 16 children who were initially suspected of benign paroxysmal vertigo in childhood [20]. In half the cases, NVCC was diagnosed. In the other group, although initially suspected, no NVCC was found; instead, a narrowing of the IAC was present. The diagnosis required radiological analysis of the shape and diameters of the IAC in the axial and coronal planes. In both groups, treatment with sodium channel-blocking drugs was effective not only in relieving dizziness but also in normalizing electrophysiological findings [20]. In the case of a narrowed IAC, we assumed that the cochleovestibular symptoms could be related to local entrapment-type neuropathy similar to carpal tunnel pathology or to radiculopathies caused by local compression [41,42]. Congenital IAC stenosis is often associated with hypoplastic CVN [43,44], but in the present study group, radiologic evaluation confirmed sufficient nerve development. We suggested that CVN compression, secondary to a small diameter of the IAC, might lead to local nerve damage, a local cranial nerve neuropathy, or even a possible ectopic excitation or inhibition of the involved cranial nerve fibers, similar or close to the NVCC [19]. The compression syndrome in the case of an NNIAC corresponds to a conflict between two anatomical elements: the content (normal-sized CVN) and the container (IAC of smaller dimensions and, therefore, smaller volume). The content alteration could be sequential: progressive compression of CVN neuronal tissue in the REZ first leads to hyperactivity; subsequently, symptoms of hypoactivity and, finally, total loss of function may occur [3]. The radiological steps and criteria that we consider relevant for a positive diagnosis of NNIAC are described here. Regarding the pathophysiologic basis of this clinical entity, we assume that a more or less extensive CVN neuropathy would be related to NNIAC. We have deliberately excluded subjects in whom a classic NVCC, according to widely accepted criteria, could be associated with an NNIAC.

We have already reported two clinical cases of “atypical” VP. The first concerned a patient with an NVCC involving the AICA and the CVN within an IAC narrowed by the presence of an osteoma [12]. This patient had been treated with poor results for years for Menière’s disease. Prescription of an antiepileptic drug provided complete relief of the symptoms suggestive of VP. We stressed the importance of considering NVCC in its two facets: the size of the IAC and the presence of a vascular loop in the IAC. The second case involved a young woman with no otologic history complaining of paroxysmal vertigo and right pulsatile tinnitus shortly after her first scuba-diving session. Cranial HRCT showed bilateral hyperpneumatization of the petrosal air cell system. MRI showed an elongated CVN in an IAC narrowed by hyperpneumatization, especially on the right side. This patient was also relieved by the prescription of low doses of antiepileptic drugs [45]. However, for the purpose of this study, the patients mentioned above and some similar cases were excluded. This is because a difference should be made between a “true” NNIAC that appears as constitutional and the secondary narrowing of the IAC that is due to other causes such as osteomas, hyperostosis of IAC, hyperpneumatization or fibrous dysplasia of the petrous bone. 

### 4.2. Further Clinical Considerations 

Although symptoms similar to those of VP may occur in the case of IAC narrowing, they seem to be more attenuated, generally less “paroxysmal” than in the classic presentation. Neuropathy in NVCC is secondary to ephapse, generated by intermittent compression through pulsations of a soft (vascular) structure. It may be different from neuropathy by compression through a hard plane such as bone. This is why, in our opinion, symptoms are more likely to be continuous. In addition, physical effort would generate more dizziness in the case of NNIAC than in the case of a “true” VP by NVCC. Short-lasting positional vertigo without latency and dizziness accentuated by head movements appeared to be the most frequent symptoms (Table 2). Most subjects had already consulted other ENT specialists and/or neurologists and were often diagnosed with “atypical” BPPV (39/59 subjects). 

From the presented data, it appears that symptoms of this pathology can appear at any age, but especially in women and predominantly in the fifth decade of life (Figure 3). As positional vertigo was a very common complaint, this led to multiple manipulations for BPPV by vestibular therapists in 30 out of 39 subjects, with mediocre or poor long-term results despite some transient improvements. Similarly, the onset of hearing loss, sometimes present on the low frequencies, could indicate endolymphatic hydrops; however, the tinnitus reported during the crisis was rather estimated on the high auditory frequencies, which is not compatible with early Menière’s disease. IACs in the SG tend to be narrowed bilaterally; however, although subjects often present with predominantly unilateral symptoms, they may occasionally complain of bilateral auditory signs, as the integrity of the CVN nerves may be affected to a different degree. As with the vestibular and audiological assessment of VP in patients with NNIAC, they may present with mild to moderate signs of unilateral hypofunction during attacks. Since vertigo attacks were rarely accompanied by unilateral audiological symptoms, identification of the affected ear based on clinical criteria was exceptional. 

Indeed, many subjects presented with a slight sensation of ear fullness and/or otalgia, which could also be confused with early Menière’s disease. This symptom could also be explained by localized suffering of the intermediate facial nerve generated by a “too” intimate contact with the IAC walls, which would thus lead to an intermediate neuralgia equivalent or very similar to what has already been described in the literature [46] and in the IHS classification (ICHD-3) (https://ichd-3.org/13-painful-cranial-neuropathies-and-other-facial-pains/, accessed on 7 December 2023). In support of this hypothesis, it can be argued that these subjects were improved by the same antiepileptic treatment, the usual analgesics being ineffective. Moreover, acetylleucine (Tanganil^®^, the most widely used anti-vertiginous in France for peripheral vertigo) has proved ineffective in treating vertigo attacks compared to carbamazepine or its derivatives.

As only small case series and clinical cases have been published [4,6,13] despite the current precise diagnostic criteria, the lack of large epidemiological data on VP in adults has recently been highlighted [7]. This partly explains the persistent controversies regarding the association of various heterogeneous audio-vestibular symptoms and the pathologically compressive vascular loops in the pontocerebellar angle and/or in the IAC [7]. Compression of the rostroventral part of the CVN would correlate with vertigo symptoms, while compression of the caudal side (cochlear nerve) could be associated with tinnitus symptoms [47,48]. It may be speculated that the clinical variability in VPs could also arise from an NNIAC in the absence of NVCC. A petrosal bone close examination in the coronal plane of the IAC could reveal a narrowing of the IAC in the medial third of the IAC, as seen in our series (Figure 1(II) and Figure 2(II)).

### 4.3. Morphometric Considerations 

We have previously emphasized the importance of protocolizing the assessment of IAC, as in the case of lumbar spinal stenosis [49], to strengthen radiological practice and create a unified view of the analysis and reporting system [20]. In the case of VP (and/or its auditory equivalents), neither neurotological clinicians nor radiologists are accustomed to carefully examining the size, shape, and dimensions of a potentially narrowed IAC, especially in the coronal plane [20]. If VP is suspected, brain MRI slices with sections centered on the IAC should be used to investigate not only contact but also compression or deformation generated by the respective vascular structure and the CVN, using axial and coronal slices. Herein, we demonstrate the value of analyzing the trajectory and anatomical relationships between these structures on coronal sections. Indeed, the shape of the IAC is not unique and can take several forms [28]. Radiological measurements, including aperture width, longitudinal length, and vertical diameter, may be sufficient when the IAC’s shape is cylindrical. However, when the IAC is conical or button shaped, these conventional measurements may be insufficient. To avoid NNIAC diagnostic failure, we chose to include additionally the measurement of the smallest diameter of the IAC in both the axial and coronal planes. As for possible future research, it would be interesting to tractography studies of the audio-vestibular pathways to try to make a correlation between the incidence of NNIAC symptoms, which, in the present report, are more vestibular than auditory (see Table 1 and Table 2) and the precisely radiological location of the area of IAC’s narrowing. Furthermore, a possible explanation of the predominance of vestibular phenomena in the case of NNIAC could be due to the fact that the vestibular contingent of the VIII nerve has fewer fibers than the auditory contingent [50,51]. Therefore, the former would be more vulnerable.

### 4.4. Limitations and Further Research

In the present study, we report a distinct possible etiology that can be considered as the cause of VP in adult subjects. To avoid confusion in the future regarding the notion of “narrowed IAC”, it would probably be interesting to develop a score (or classification) of the degree of narrowing of this structure based on anatomical-clinical criteria possibly correlated with audio-vestibular symptoms. In the meantime, we preferred the expression NNIAC throughout this manuscript precisely to differentiate it from the classically defined entity proposed by Valvassori and other authors. The results of this study must also be viewed considering certain limitations. The sample size was quite small, so it was insufficient to obtain statistical data and to be generalized to the whole population. Outcome estimates are based on retrospective observational data and are, therefore, subject to bias. In addition, therapeutic data were sparse, so there is no clear judgment on therapeutic response. The efficacy of antiepileptic drugs is an important criterion for the diagnosis of VP secondary to the presence of a vascular loop in the pontocerebellar angle [7]. We have already reported arguments in favor of the efficacy of antiepileptic treatments for VP or equivalence secondary to the presence of an NNIAC in children [20]. 

Finally, two identical control subject groups of cochlear implant patients were used to compare IAC sizes among the three subpopulations. Given the retrospective and non-randomized nature of this study, on the one hand, it was not possible to obtain all radiological measurements in the control adult group (CG1) composed of subjects with no vestibular complaints; on the other hand, the CG2 subjects underwent all radiological and vestibular examinations as a part of their pre- and post-operative protocol for cochlear implanted candidates. 

## 5. Conclusions

In the present study, we report a clinical entity that reasonably appears to be responsible for a similar, but not identical, clinical presentation to VP rather than associated with the presence of an NNIAC. Its audio-vestibular features, as well as its radiological findings, are, in our opinion, sufficiently specific to justify considering it an independent neurotologic entity. At the same time, this allows us to differentiate it from the “true” narrowed IAC previously introduced by Valvassori. Diagnosis requires a careful analysis of the shape and diameters of the IAC. MRI radiologic measurements only in the axial plane seem to be insufficient to confirm the diagnosis, and, therefore, an analysis of diameters in the coronal plane is highly recommended.

## Figures and Tables

**Figure 1 jcm-12-07580-f001:**
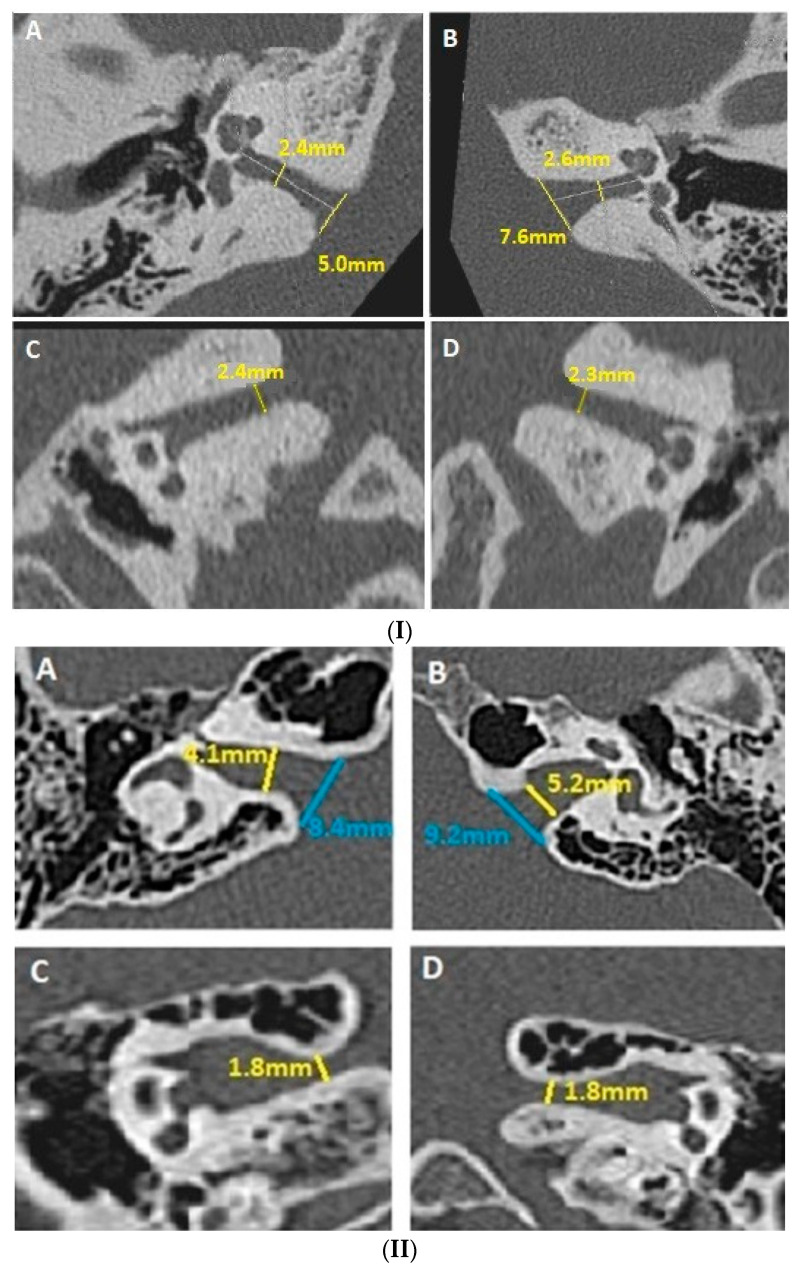
(**I**) HRCT of the temporal bone; cylindrical-shaped IAC. Assessment of the smallest anteroposterior and craniocaudal IAC diameters in both axial (**A**,**B**) and coronal planes (**C**,**D**). (**II**) HRCT of the temporal bone; button-shaped IAC. Assessment of the smallest anteroposterior and craniocaudal IAC diameters in both axial (**A**,**B**) and coronal planes (**C**,**D**). In this example case, if the measurements had been limited only to the axial plane (**I**(**A**,**B**)), the diagnosis would have been missed because the IAC is narrowed only in the coronal plane (**I**(**C**,**D**)). HRCT: high-resolution computed tomodensitometry; IAC: internal auditory canal.

**Figure 2 jcm-12-07580-f002:**
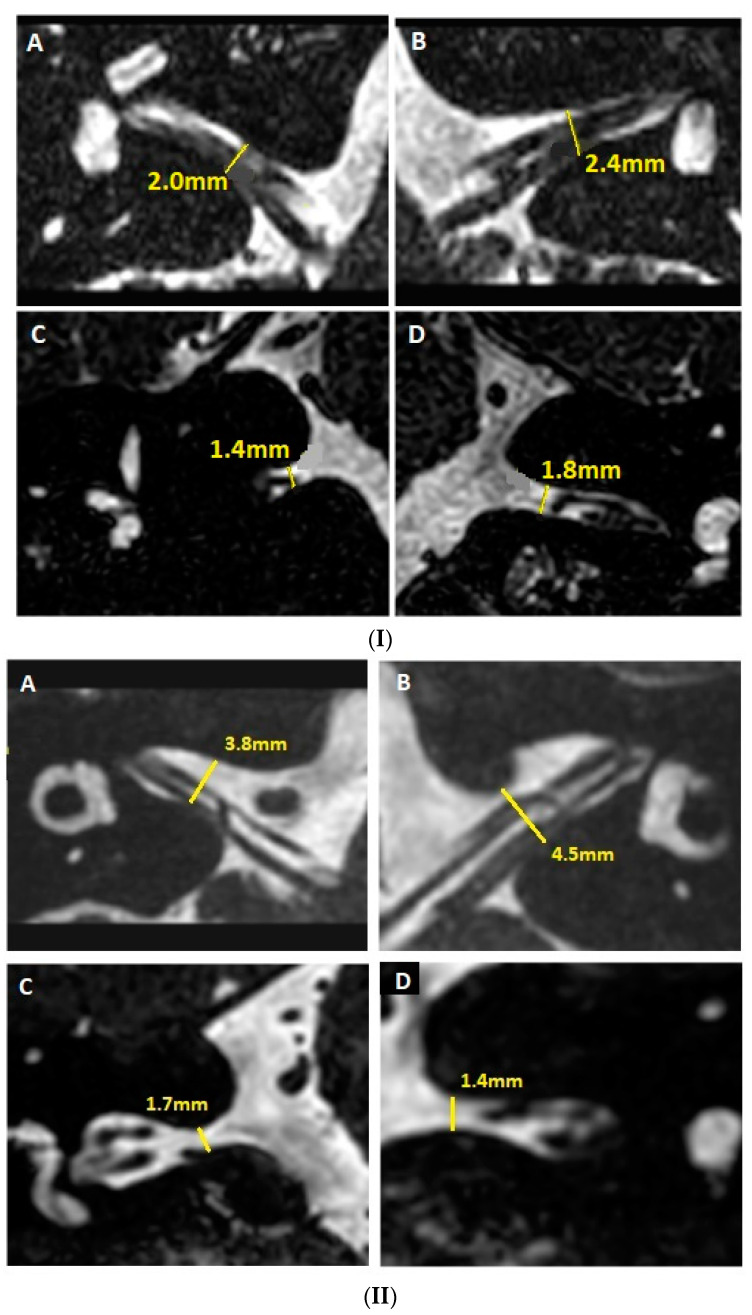
(**I**) MRI of the IAC; cylindrical-shaped. Evaluation of the smallest anteroposterior (**A**,**B**) and craniocaudal (**C**,**D**) diameters of the IAC. (**II**) MRI of the IAC; button shaped. Evaluation of the smallest anteroposterior (**A**,**B**) and craniocaudal (**C**,**D**) diameters of the IAC. MRI: magnetic resonance imaging; IAC: internal auditory canal.

**Figure 3 jcm-12-07580-f003:**
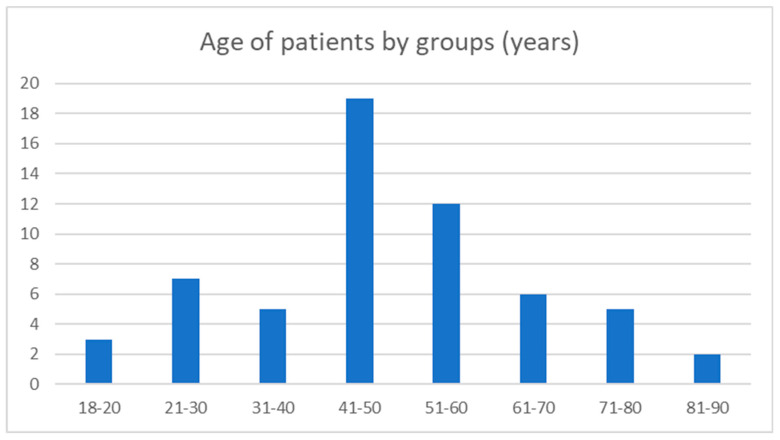
Groups on patients by age; the higher incidence of diagnosis was the 5th decade of life.

**Figure 4 jcm-12-07580-f004:**
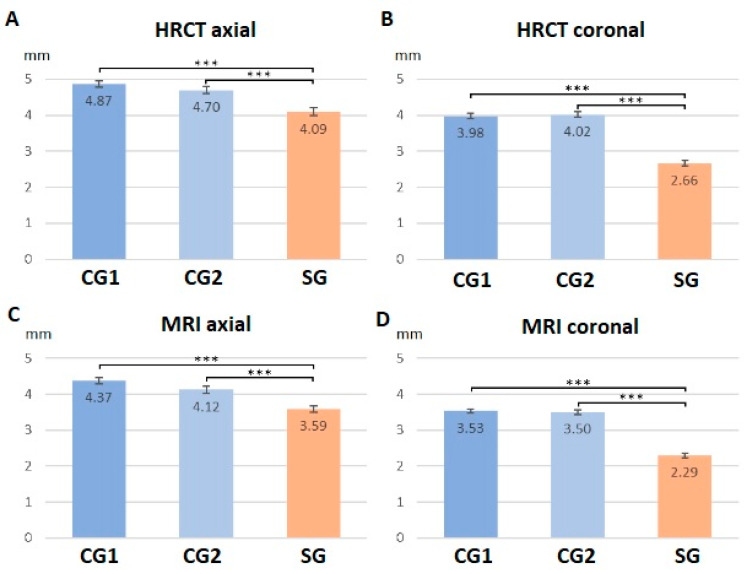
Comparison between mean values, in mm, of the IAC dimensions in axial (**A**,**C**) or coronal (**B**,**D**) planes, measured using HRCT (**A**,**B**) or MRI (**C**,**D**) for each group. Error bars represent the standard error of the means; significance of differences with *p* < 0.001 is indicated using ***. CG1: control group 1 (adults); CG2: control group 2 (children); SG: study group.

**Figure 5 jcm-12-07580-f005:**
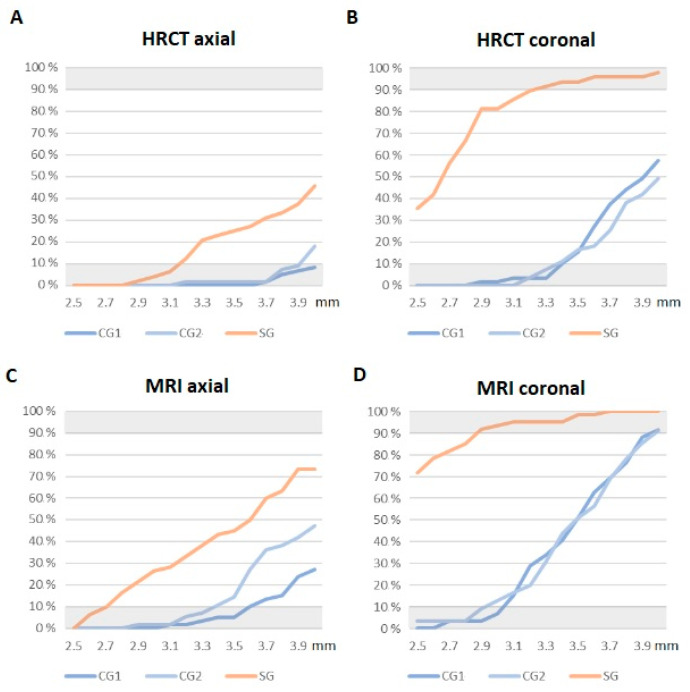
For each threshold value in mm (*x*-axis), the percentage of subjects above these values (*y*-axis); percentage measured for each variable: axial HRCT (**A**), coronal HRCT (**B**), axial MRI (**C**), and coronal MRI (**D**), and for each group: CG1: control group 1 (adults); CG2: control group 2 (children); SG: study group.

**Table 1 jcm-12-07580-t001:** Clinical auditory signs in the study group. SNHL: sensori-neural hearing loss.

	Unilateral (*n*=)	Bilateral (*n*=)
Hyperacusis	5 (8.5%)	17 (28.8%)
Tinnitus during physical effort	5 (8.5%)	15 (25.4%)
Continuous tinnitus	1 (1.7%)	8 (13.5%)
Otalgia	10 (16.9%)	0 (0%)
Progressive non-genetic SNHL	4 (6.8%)	7 (11.9%)
Ear fullness	5 (8.5%)	0 (0%)

**Table 2 jcm-12-07580-t002:** Clinical vestibular signs in the study group and previously evoked diagnosis.

	Number of Subjects (Percentage of the 59 Subjects in the Study Group)
Positional vertigo	39 (66%)
Acetylleucine ineffectiveness	29 (49%)
Dizziness on rapid head movements	26 (44%)
Exercise-induced vertigo	25 (42%)
Chronic imbalance and dizziness	23 (38.5%)
Acute vertigo	3 (5%)
Progressive bilateral vestibulopathy	2 (3.3%)
Paroxysmal vertigo	2 (3.3%)

## Data Availability

The data presented in this study are available on request from the corresponding author. The data are not publicly available due to ethical, legal, and privacy issues.

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
