# Peer review of "The “Near”-Narrowed Internal Auditory Canal Syndrome in Adults: Clinical Aspects, Audio-Vestibular Findings, and Radiological Criteria for Diagnosis"

_jcm, 2023, doi:10.3390/jcm12247580_

Round 1

Reviewer 1 Report

Comments and Suggestions for Authors

In this article, Eugen Ionescu and colleagues tried to explore possible alternative pathophysiological interpretation in patients vith symtoms suggestive of vestibular paroxysmia, but no MRI evidence of neurovscular conflict. To do this, they focused on internal acoustic canal size.

The article is interesting and, methodology is sound. However, some items need to be addressed

- when defining the control groups (based on adults and children with profound deafness, treated with C.I.), Authors should also clarify if patients had inner ear malformations

- in the discussion section, other causes of IAC stenosis shuold be mentioned and discussed (e.g. IAC osteomas)

- English language should be revised for grammar and style

Comments on the Quality of English Language

Overall, English language is fair, however the manuscript should be revised for grammar and style

Author Response

Dear Reviewer,

Thank you for your relevant comments.
Even if we made a reference in the discussions to several cases of narrowed IAC of the secondary type, it is true that we did not clearly specify the difference
between the "essential" and "secondary" N IAC.
I have fixed this lack now in the actual version of the manuscript  - page 11 /line 360 - 364. I also clearly stated, adding the necessary, that all control subjects (CG1 and CG2) had strictly normal imagery by HRCT and MRI of the inner ear and the petrous bone. The quality of written English was also revised.

Reviewer 2 Report

Comments and Suggestions for Authors

The article suggests an original idea by proposing that certain manifestations of a VP may be linked to pathological changes in the diameter of the internal auditory canal. This idea could have significant implications for the approach to patients with vestibular disorders. 

The authors used two control groups – one with adults and one with children – who underwent imaging for cochlear implants and did not have vestibular pathology. The two control groups should be defined more precisely both in terms of number and criteria for inclusion in the group.

The discussion and conclusions support the imaging findings and the pathological changes identified. Figures and tables are well-made and suggestive and they enhance the overall presentation of the article.

Suggestions for the authors:

The two control groups should be defined more precisely both in terms of number and criteria for inclusion in the group.

Author Response

Thank you very much for the reviewer's appreciation.
We corrected the lack of more precise information where needed for the number of subjects included in the study groups as you suggested cf. page 3 /line 109 and page 7 - lines 207, 211, 213
and 214

Reviewer 3 Report

Comments and Suggestions for Authors

The authors suggested that the patients with near narrowed IAC may manifest clinical features similar to vestibular paroxysmia.

Although fairly interesting and novel, this study did not investigate the status of CNV within the IAC or the presence of NVCC, which would be more correlated to the clinical manifestation. At least, the authors are recommended to explain the relation of each vestibular and auditory symptoms with radiologic findings.

Author Response

Thanks to the Reviewer for the appreciation. As our primary goal in this work was
to sensitize fellow neurologists about the possibility of this pathology, we did
not provide in this initial report studies on tractography of the VIIIth nerve and possible correlation with some symptoms -
as is otherwise possible in our days, it is true (Shapey J, Vos SB, Vercauteren T, Bradford R, Saeed SR, Bisdas S, Ourselin S. Clinical Applications for Diffusion MRI and Tractography of Cranial Nerves Within the Posterior Fossa: A Systematic Review. Front Neurosci. 2019 Feb 7;13:23. doi: 10.3389/fnins.2019.00023. PMID: 30809109; PMCID: PMC6380197)
As the subjects in the SG were gathered progressively, quite disparately in time - almost 7 years - this did not allow us to design the study as we would have liked - so we are only in a retrospective clinical research phase. In the future, however, this work will definitely have to be done - just as I added this in the manuscript as a sign of your demand and a possible explanation for the prepondereance of vestibular symptoms in case of NNIAC .-pages 12&13 in Discussions - line 426-433

Round 2

Reviewer 3 Report

Comments and Suggestions for Authors

The manuscript has been substantially improved after revision.